# Novelties in 3D Transthoracic Echocardiography

**DOI:** 10.3390/jcm10030408

**Published:** 2021-01-21

**Authors:** Gianpiero Italiano, Laura Fusini, Valentina Mantegazza, Gloria Tamborini, Manuela Muratori, Sarah Ghulam Ali, Marco Penso, Anna Garlaschè, Paola Gripari, Mauro Pepi

**Affiliations:** Centro Cardiologico Monzino IRCCS, Via Parea 4, 20138 Milan, Italy; laura.fusini@ccfm.it (L.F.); valentina.mantegazza@ccfm.it (V.M.); gloria.tamborini@ccfm.it (G.T.); manuela.muratori@ccfm.it (M.M.); sarah.ghulamali@ccfm.it (S.G.A.); marco.penso@ccfm.it (M.P.); anna.garlasche@ccfm.it (A.G.); paola.gripari@ccfm.it (P.G.); mauro.pepi@ccfm.it (M.P.)

**Keywords:** 3D echocardiography, transillumination, machine learning

## Abstract

Cardiovascular imaging is developing at a rapid pace and the newer modalities, in particular three-dimensional echocardiography, allow better analysis of heart structures. Identifying valve lesions and grading their severity represents crucial information and nowadays is strengthened by the introduction of new software, such as transillumination, which provide detailed morphology descriptions. Chambers quantification has never been so rapid and accurate: machine learning algorithms generate automated volume measurements, including left ventricular systolic and diastolic function, which is extremely important for clinical decisions. This review provides an overview of the latest innovations in the echocardiography field, and is helpful by providing a better insight into heart diseases.

## 1. Introduction

Since its introduction into clinical practice in early 2000s, three-dimensional (3D) echocardiographic imaging has represented a major innovation in cardiovascular ultrasound. Like computed tomography (CT) and magnetic resonance imaging (MRI), 3D echocardiography allows the acquisition of real-time 3D data sets with adequate spatial and temporal resolutions. Image cropping and rotation with infinite cut planes, both on- and off-line, generate anatomically orientated views, which are extremely useful in various clinical settings.

The areas of application of 3D echocardiography have gradually expanded: the analysis of geometry and function of the heart chambers [1], the evaluation of the results of transcatheter procedures [2], and the new ways of visualizing heart valves [3] are revolutionizing clinical day practice.

This review describes the newer modalities of transthoracic (TTE) 3D echocardiography that may enhance valve disease characterization and heart chamber volume quantification as well.

## 2. Assessment of Valve Pathologies 

Echocardiography is the method of choice to evaluate patients with valve disease. It allows the accurate assessment of severity of valve lesions, etiology, mechanisms and anatomic lesions, left ventricle (LV) and left atrial (LA) remodeling, right chamber dimensions and function and consequently defines the indication and the probability of successful of percutaneous or surgical procedures. 

Recent guidelines stated that real-time three-dimensional (RT3D) imaging modalities are useful in the presentation of realistic views of heart valves and are ideal to investigate the anatomy and function of each of the heart valves [1]. 

### 2.1. Mitral Valve

In mitral valve (MV) disease, 3D imaging enhance the evaluation of all anatomic and functional details, including subcomponents of the MV apparatus (annulus, leaflets, chordae and papillary muscles). Both qualitative and quantitative evaluations of degenerative MV disease have been substantially improved by 3D echocardiography.

In 2012, ESC guidelines [1] stated that “3DTTE and TEE assessments of mitral valve pathology should be incorporated into routine clinical practice”. This statement was mainly based on 3D transesophageal (TEE) data. However, in 2006 it was reported that both 3DTTE and TEE were feasible and useful methods to identify the location of MV prolapse [4]. Three-dimensional imaging reconstructions provided a very full description of valve pathology in comparison with 2D techniques and represented an important added value for MV repair planning [5]. Since then, transducers, software and hardware have improved dramatically in terms of acquisition, temporal and spatial resolution and image rendering. Several studies have confirmed the superiority of 3DTTE in the evaluation of the MV apparatus as well as of the aortic (AV) and tricuspid valve (TV) [6,7,8,9].

One of the main indications for 3DTTE is the morphologic description of MV leaflets in several diseases, mainly MV regurgitation and MV stenosis. The first studies in the early 2000s were concentrated on the planimetry of the valve in MV stenosis [10] and in the identification of scallop lesions in MV prolapse [1,4,5,11]. Using 3D analysis software, nowadays incorporated on-line in the majority of ultrasound units, it is possible to perform a multiplane reconstruction from the dataset and generate a perfectly aligned cross-sectional image of the MV at the leaflet tips. Considering stenotic MV almost like a funnel-shaped structure, the valve area is at its minimum at that level and it represents the true anatomical MV area [12].

The evaluation of MV morphology and identification of scallop lesion is fundamental in patients with MV prolapse. In a routine TTE, a comprehensive evaluation contributes to the diagnosis, prognosis and surgical timing of severe degenerative MV regurgitation: heart chamber dimensions and function, detailed MV morphology, site and severity of the regurgitation, pulmonary systolic pressure [13,14]. In this regard, 3DTTE concur in the evaluation of heart chamber volume and function in MV disease. Several papers showed that 3D echocardiography is superior in comparison with the corresponding 2D techniques in the description of MV pathology. In particular, since RT3D has a similar accuracy of 2DTEE, this technique may be integrated in the standard 2D examination and should be regarded as an important tool in the decision process for MV repair [1,4]. Gutierrez et al., with the first commercial RT3D transducer, emphasized that segmental analysis in mitral prolapse can be performed as accurately as 2DTEE [6]. False negatives tend to appear around the anterolateral commissure, whereas false positives tend to appear around the posteromedial commissure. The highest accuracy was reached in central scallops. Tamborini et al. [11] not only confirmed these data but also demonstrated that RT3D TTE is a feasible, not time-consuming, non-invasive useful method in order to identify the location of simple and complex lesions in patients with MV prolapse undergoing MV repair. Therefore, this technique should be regarded as an important adjunct to standard 2D examination in recognizing all the components of MV leaflets and may facilitate the prediction of surgical procedures. This has been very recently confirmed, showing that a pre-operative 3DTTE may also predict whether cases with MV prolapse will undergo simple or complex surgical repair procedures, with a potential impact on long-term residual regurgitation [15].

Moreover, 3DTTE diagnostic accuracy is still improving thanks to new transducers, software and transillumination (TI) techniques [16]. Comprehensive mitral valve reconstructions help to identify chordal rupture and cleft recognition as well as scallop identification [17]. TI is a new tool that improves visualization of cardiac structures due to shadow effects achieved by using a virtual light source. This method is user friendly: a touch screen allows operators to change the light source orientation, creating a 3D reconstruction that markedly emphasize a shadow effect. Thus, the most relevant innovation is the perception of depth, which is notably improved with its use. TI and the new transparency mode may help to clearly delineate pathologic areas such as prolapsed valve leaflet scallops empowering a clear distinction of cardiac and extra-cardiac structures (Figure 1).

Starting from the morphology and dimensions of the valve annulus, 3DTTE has been proposed as an ideal method to analyze the mitral, tricuspid and aortic annula. The MV annulus has been studied in patients with organic or functional mitral regurgitation and after surgical and percutaneous procedures [18,19,20]. 

### 2.2. Tricuspid Valve

The evaluation of TV leaflets is limited by the standard 2DTTE. It is rarely possible to simultaneously visualize the three leaflets and their recognition needs a very accurate and comprehensive evaluation from different views. By contrast, 3DTTE is generally easy and has the unique capability of obtaining a simultaneous visualization of the three leaflets, commissures and their attachment to the annulus. Badano et al. demonstrated the clinical utility of 3DTTE in the assessment of functional TV regurgitation [8,9], combining 3D data on valve morphology and dynamics, TV annulus and 3D regurgitant jet evaluation. Functional regurgitation of the TV may also be related to cardiac implantable electronic device (lead-induced regurgitation). Specifically, while a correct commissure position of the lead does not interfere with leaflet motion, lead interference may cause severe regurgitation due to adherent, impinging or entangled mechanisms [21]. Recently it has been demonstrated that 2DTTE and 2DTEE measurements underestimated TV annulus dimensions in all echo views compared to 3D values, irrespective of the 3D method [22]. No differences were found between 3DTTE and 3DTEE and, for practical reasons, 3DTTE may be proposed as an ideal method to measure the TV annulus, particularly in patients undergoing MV surgery. In fact, all structures with an oval shape, or with the typical saddle shape, may be better examined with a 3D approach. A quantitative 3D evaluation of the annulus (major and minor diameters as well as annulus area) is more accurate than a 2D evaluation. From a clinical point of view, tricuspid cut-off for TV repair, in patient undergoing isolated or combined surgery (for example in MV patients), may largely benefit from a 3D-based analysis.

Several other pathologies may benefit from 3DTTE diagnosis such as tricuspid endocarditis, valve tumors, congenital and acquired pathologies. Figure 2 shows several examples of different pathologies that may benefit from 3DTTE evaluation, and Figure 3 displays the potential use in a case of the bicuspid aortic valve.

### 2.3. Aortic Valve

Although quantitatively less relevant than the contribution offered for evaluation of morphology and function of atrioventricular valves, 3D echocardiography, especially from the transesophageal approach, has become critical for evaluation of the anatomy of the aortic valve [23]. In aortic stenosis it is mainly used for correct characterization of the LV outflow tract, which is crucial for estimating the prosthesis size for transcatheter valve implantation procedures [24]. Several articles showed that also 3DTTE, besides 3DTEE and CT, is accurate in the measurements of this oval shape structure [25,26]. 

In aortic insufficiency, 3DTTE is used to define the anatomy of the valve, the mechanism of insufficiency, and to indicate the need for eventual reconstructive surgery [27].

### 2.4. Grading the Severity of Valve Lesions

Accurate quantification of mitral, aortic and tricuspid regurgitation is important for decisions regarding surgery and predicting risk. Current guidelines [28] propose integration of specific, supportive, and quantitative echocardiographic features to classify the severity of MR and other valves. These methods include vena contracta width (VCW), regurgitant volume (RVol) and fraction (RF), and effective regurgitant orifice area (EROA). Although these quantitative techniques can be accurate and reproducible in single centers, there can be significant interobserver variability among centers.

Recent technological advances in 3DTTE and 3DTEE have provided new tools for MR quantification. As concerns 3DTTE, images are generally acquired from the apical view with settings similar to the standard 2D Doppler approach. Newer ultrasound systems will allow real-time optimization of the 3D color Doppler in live 3D mode or single heart-beat full-volume mode, but the acquisition should still be gated. End-expiratory breath-hold is desirable for gated acquisitions, and all 3D volumes should be checked perpendicular to the ultrasound sweep plane to ensure that stitching artifacts are absent. 

3D VCA measurements by TEE is superior to the 2D PISA method in determination of regurgitation severity in multiple native and prosthetic valves. However, new methods for automated quantification of regurgitant jets are still under evaluation and probably will very soon be available in clinical practice [29].

In the near future, we may also foresee that the “true hole” of the regurgitant lesion may be reconstructed from 3D data sets [30]. In order to assist in a more accurate and non time-consuming approach for regurgitant quantification, automated color jet evaluation and/or quantitative 3D assessment of stroke volumes, as performed in MRI, will largely be used.

Table 1 shows comparisons among 2D and 3D modalities regarding assessment of valve morphology and function.

### 2.5. Left Ventricular Volumetric and Functional Assessment 

In left heart valve disease, LV and LA evaluation is crucial for the diagnosis, prognosis and timing of surgery or for percutaneous procedures. Indeed, both in aortic and mitral stenotic or regurgitant lesions, clinical and therapeutic decision are based not only in severity of the lesion but also on LV dimensions and systolic function and LA dimensions [13,14,28]. In MV regurgitation (MR), surgery is recommended for asymptomatic patients with chronic severe primary MR and LV dysfunction (left ventricle ejection fraction (LVEF) 30% to 60% and/or left ventricular end-systolic diameter (LVESD) > 40 mm). Asymptomatic patients with severe aortic regurgitation and an impairment of LV function (LVEF < 50%; LV diastolic diameter > 70 mm or left ventricular end-systolic diameter (LVESD) > 50 mm) go towards a worse outcome reason that surgery should be pursued when these cut-offs are reached or even earlier [31,32,33]. In aortic stenosis, surgery should be considered in symptomatic and even asymptomatic patients with low-flow, low-gradient aortic stenosis and reduced ejection fraction without contractile reserve, particularly when CT calcium scoring confirms severe aortic stenosis [34,35,36,37,38].

Therefore, as from these quoted guidelines, M-mode diameters and 2D biplane LV volumes were considered the standard method for the assessment of surgical timing. Recently 3D echocardiography, tissue Doppler and strain rate imaging have been proposed as useful techniques, particularly in patients with borderline LVEF, where they may help in the decision for surgery [39,40,41,42]. In this field 3D echocardiography moved rapidly from complex and off-line LV and LA volume calculation to semiautomated and automated methods and, in the last few years, to deep learning calculations [43,44,45,46,47,48]. In valve diseases, LA volume measurements are very useful, and LA size and function parameters are associated with adverse outcomes. Moreover, recent data suggest that phasic LA function also hold prognostic information [49,50,51,52]. 3DTTE provides more accurate and reproducible quantification of LA volumes than 2DTTE when compared with CMR reference standards [53]. New machine learning 3D methods implemented in commercial ultrasound units allow accurate and rapid measurements of LA volume and function [54]. Correlation between automated volumes and CMR are excellent with minimal bias [55].

Moreover, after launching the acquired 3DE dataset (after a few seconds’ acquisition from the 4-chamber view), the examiner has immediately (about 30 s) automated chambers assessment on the display. LA and LV contours on 4-, 3-, and 2-chamber cut-planes are extracted from the 3DE datasets, providing the ideal recognition of the 2 cavities and LV/LA reciprocal volumetric changes throughout the cardiac cycle. All measurements are also included in the display (LVEF, LA and LV volumes and function). The different phases of the cardiac cycle are identified using maximum and minimum values of the volume curves and their first time-derivative LV and LA curves [56]. In the case of improper borders detection, manual corrections are easily performed [57]. Finally, 3D echocardiography analysis of LV mass using novel ML-based algorithm is feasible, fast, and accurate and may thus facilitate the incorporation of 3DE measurements of LV mass into clinical practice [58].

Thus, a comprehensive TTE may include not only all 2D echo and Doppler parameters but also 3D evaluation of valve morphology and 3D measurements of left heart chambers [59]. Figure 4 shows echo findings in a patient with severe MV regurgitation in whom 3D automated calculations of LV and LA volume and function have been applied. 

### 2.6. Right Ventricular Assessment

Right ventricular (RV) dimensions and function are known to be of clinical importance in many cardiac diseases. The 3DTTE determined RV ejection fraction (RVEF) was independently associated with cardiac outcomes in patients with diverse backgrounds including valve disease [60]. RVEF offered incremental value over clinical risk factors and the other echocardiographic parameters including LV systolic and diastolic function for predicting future adverse outcome. Nowadays, the normal reference values of RV volume and function by 3DTTE have been established [61]. Several papers assessed the feasibility, accuracy and additional clinical value over 2DTTE in different valve diseases [62,63,64,65,66,67,68]. In clinical practice, 3DTTE with new automated software may easily acquire in a few seconds images from an adapted 4-chamber view and can obtain volumes, RVEF, global and free wall RV strain, TAPSE and fractional area change [69].

We recently demonstrated that patients undergoing MV surgery and TV annuloplasty with an increased TV annulus dimensions and a marked right chambers remodeling, probably had an advanced stage of the disease [70]. The 3D echocardiography may better define the right chamber’s overload and RV function as well as TV annulus morphology and may be proposed as an ideal method to quantify parameters useful for define timing of TV annuloplasty. This is also crucial in severe tricuspid regurgitation, both primary and secondary to left heart valve disease. In these cases, severity of the TV lesion should be correlated to the TR annulus dimensions and right atrium and RV volumes. As previously discussed in annulus and leaflet morphology sections, 3DTTE may better cover TV characteristic adding also hemodynamic consequences of the disease in terms of RV volumes and function (Figure 5).

Table 2 shows comparisons among 2D and 3D modalities regarding cardiac chamber quantification and function.

## 3. Conclusions

Imaging of the heart and related structures has progressively evolved over the last few years. Three-dimensional (3D) echocardiography has come a long way from its debut and represents the most useful innovation in cardiovascular ultrasound. Its use can be helpful for gaining a better insight into valve diseases and for rapid and accurate volumes and function assessment. Further improvements of the image quality and resolution associated with the use of artificial intelligence techniques, for automated quantitative analysis of the data, will allow 3DTTE to reduce analysis time and make quantitative assessments more objective and reproducible. 

## Figures and Tables

**Figure 1 jcm-10-00408-f001:**
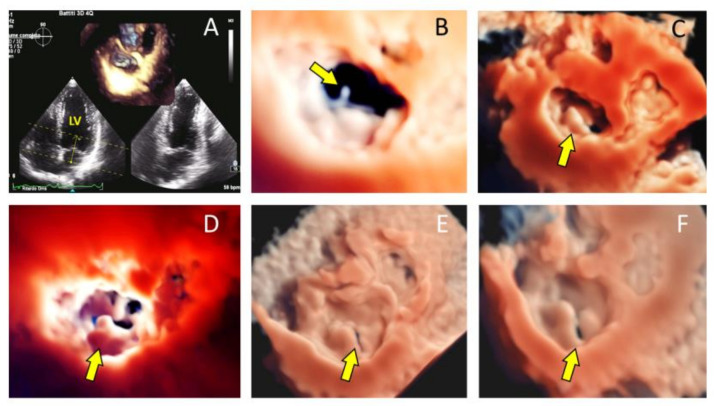
Transillumination techniques in mitral valve prolapse. (**A**): Four-chamber view showing (arrow and area of interest included in the 2 yellow lines) how to visualize the surgical view of the MV; (**B**): example of visualization of a chordal rupture (arrow) from the surgical view; (**C**): loculated P2 prolapse (arrow); (**D**): large P2 prolapse (arrow); (**E**,**F**) early- and mid- systolic frames of a P2 prolapse associated with a cleft P2/P3 (arrow).

**Figure 2 jcm-10-00408-f002:**
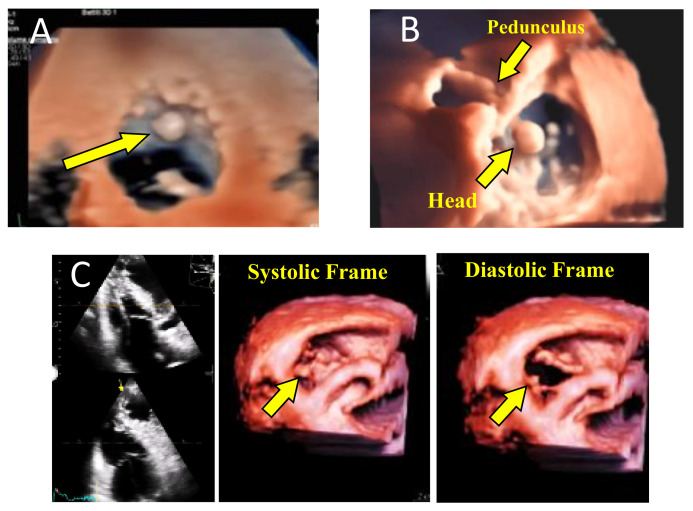
Three examples of cases in whom transthoracic 3D echocardiography (3DTTE) was particularly useful: (**A**) left ventricle (LV) myxoma, (**B**) Tricuspid valve fibroelastoma and (**C**) precise localization of a pacemaker lead crossing the tricuspid valve.

**Figure 3 jcm-10-00408-f003:**
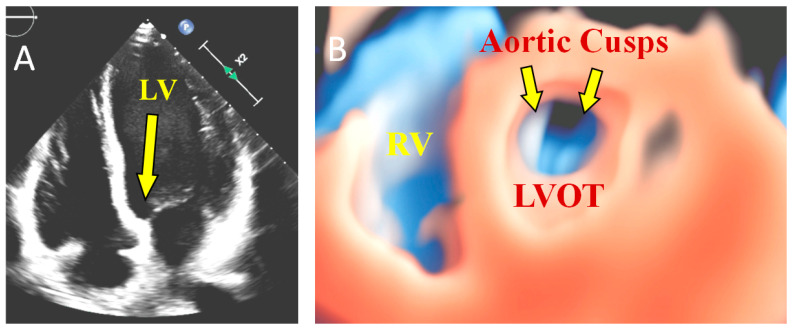
Bicuspid aortic valve 3D reconstruction: (**A**) the arrow shows the perspective view to visualize the LV outflow tract; (**B**) transillumination technique of LVOT and aortic cusps. LV: left ventricle; LVOT: left ventricle outflow tract; RV: right ventricle.

**Figure 4 jcm-10-00408-f004:**
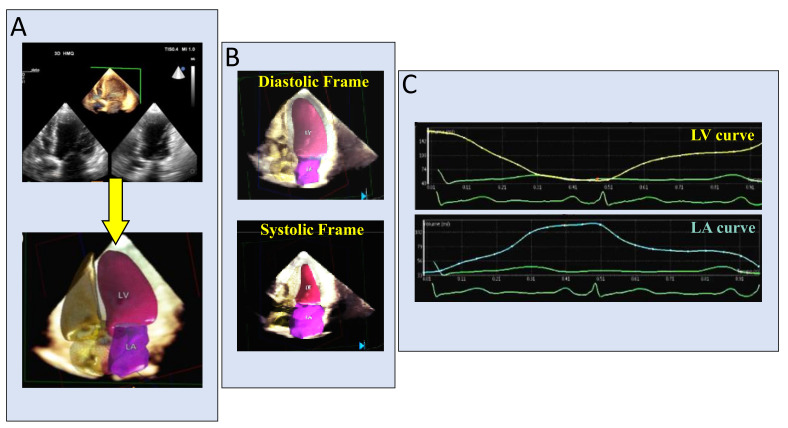
Dynamic automated evaluation of the LV and LA volumes and function. After launching the acquired 3DE dataset (**A**), the examiner has immediately on the display LV and LA with automated recognition of the 2 cavities (**B**) and LV/LA reciprocal volumetric changes, throughout the cardiac cycle, with all measurements and functional curves (**C**).

**Figure 5 jcm-10-00408-f005:**
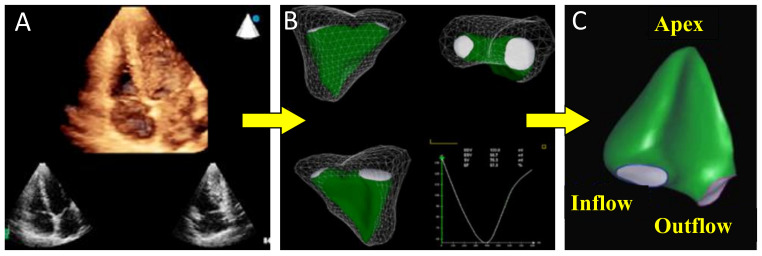
Acquisition and reconstruction of the right ventricle (RV). (**A**): adapted 4-chamber view for acquisition of the entire RV cavity; (**B**): reconstruction on the display of the RV cast (cine loops) and all measurements (RV volumes and function). (**C**): magnification of the RV cast and morphology of inflow, outflow and apex of the RV.

**Table 1 jcm-10-00408-t001:** Head to head comparison of 2D and 3D modalities in valve morphology assessment.

	2DTTE	2DTEE	3DTTE (Standard)	3DTTE (Advanced *)	3DTEE
Feasibility (all valves)	+++	++++	+++	++++	++++
Training and communication	++	++	+++	++++	++++
Accuracy in MV morphology assessment vs. surgical inspection	++	+++	+++	+++	++++
Accuracy in MV cleft and chordal rupture assessment vs. surgical inspection	++	+++	++	+++	++++
Accuracy in TV morphology assessment	++	++	+++	++++	+++
Accuracy in annulus quantification (Ao, MV and TV)	++	++	+++	++++	++++
MV and TV regurgitant jet quantification	+++	+++	+++	+++	++++
Accuracy in Ao morphology assessment	++	++++	++	+++	++++
Usefulness in percutaneous procedure assessment	+	+++	++	++	++++

TTE = transthoracic echocardiography; TEE = transesophageal echocardiography; MV = mitral valve; TV = tricuspid valve; Ao = aortic valve; * advanced 3DTTE includes transillumination and transparency. + = scale of peculiar abilities.

**Table 2 jcm-10-00408-t002:** Head to head comparison of 2D and 3D modalities in the assessment of heart chambers volumes and function.

	2DTTE	2DTEE	3DTTE (Standard)	3DTTE (Advanced *)	3DTEE
Feasibility of acquisition/measurements	+++	++	+++	++++	++
Time for acquisition/measurements	++	++	+++	++++	++
Accuracy in LV volume quantification vs. MRI	++	++	+++	++++	++
Accuracy in LV Ejection fraction vs. MRI	++	+	+++	++++	+
Reproducibility of LV quantification	++	++	+++	++++	++
Accuracy in RV volume quantification vs. MRI	+	++	+++	++++	++
Accuracy in RV ejection fraction vs. MRI	+	+	+++	++++	+
Reproducibility in RV quantification	+	+	+++	++++	++
Accuracy in LA quantification vs. MRI	++	+	+++	++++	++

TTE = transthoracic echocardiography; TEE = transesophageal echocardiography; MRI = magnetic resonance imaging; LV = left ventricle; RV = right ventricle; LA = left atrium. * advanced 3DTTE includes machine-learning software. + = scale of peculiar abilities.

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
