# Peer review of "Novelties in 3D Transthoracic Echocardiography"

_jcm, 2021, doi:10.3390/jcm10030408_

Round 1
Reviewer 1 Report
The authors aimed to desribe the novelties in 3D Transthoracic Echocardiography. They concluded that imaging of the heart and related structures has progressively evolved over the last years. 3D echocardiography has represented the most useful innovation in cardiovascular ultrasound. While 3DTEE is a standard and routine examination in several settings, 3DTTE is still in search of clinical indications.
I have a few concerns related to this paper. The subject of this review is important and dynamically evolving. Unfortunately, the design of the manuscript is a bit chaotic and unclear for the reader.
- It is not clear what kind of cardiac structures are assessed in this review.
- What is the time frame of the papers included in this review?
- What kind of papers were included? Cross sectional studies? Guidelines?
- Rearranging the design by dividing it by cardiac structures seem to be benefitial for the paper.
Author Response
Please see the attachment
Response to reviewer #1. Comments
The Authors would like to express their gratitude for the favorable comments and appropriate suggestions. Revision has been made accordingly.
Points 1. 2. and 3. In this new version we tried to better define the design of the Review on 3DTTE starting from Valve disease assessment and then moving to Heart Chambers quantification. Moreover we also better reported that even though this Review covers mainly technical novelties of 3DTTE we also quoted the technical and historical background of the advancement of the method adding references. Since it is not a metanalysis of all studies in this field we tried in this new version to include original papers and the ESC Guidelines related (2012; no other Guidelines has been later on produced)
Point 4. Thanks to this important suggestion; we rearrange the paper and we also inserted as also suggested by Reviewer 2 new Tables to better define qualitative and quantitative comparisons among the different old and new 3DTTE modalities in this field.

Reviewer 2 Report
The paper outlines the advantages of 3D echo for better analysis of heart structure and function. The authors mainly review the recent developments in 3D echo acquisition and machine learning (ML) based analysis of echo data, and also provide a few visual examples.
The strengths of the paper are:
- It presents an interesting summary of 3D echo advantages in the assessment of various heart structures.
However, the main limitations of the paper are the lack of novelty and experiments. In specific:
- The authors may highlight the main points derived from their review through a quantitative or qualitative analysis. They may support their claims by presenting numerical comparisons showing the advantages of the reviewed 3D echo techniques in clinical experiments, or present a comparison of the efficacy of different existing ML-based algorithms.
Author Response
Please see the attachment
Response to reviewer #2. Comments
The Authors would like to express their gratitude for the favorable comments and appropriate suggestions. Revision has been made accordingly.
In this new version of the paper:
a. In this new version we tried to better define the design of the Review on 3DTTE starting from valve assessment and then moving to heart chambers quantification. Moreover, we also better reported that even though this Review covers mainly technical novelties of 3DTTE we also quoted the technical and historical background of the advancement of the method. Since it is not a metanalysis of all studies in this field we tried in this new version to include original papers and the 3D Guidelines (2012; no other Guidelines has been later on produced). We rearrange the paper and we also inserted 2 new Tables to better define qualitative and quantitative comparisons among the different old and new 3DTTE modalities in this field.
b. As concerns ML-based algorithms we quoted articles on this approach and reported main technical details of the method. We add more references in this regard.

Reviewer 3 Report
Italiano et al. describe the importance and value of the use of echocardiography in clinical routine. It is a well-written manuscript and presents an essential diagnostic tool. The authors should add the use of echocardiography in the interventional field (interventional valve therapies, TASH, LAA-Occluder ...).
Author Response
Please see the attachment
Response to reviewer #3. Comments
The Authors would like to express their gratitude for the favorable comments and appropriate suggestions. Revision has been made accordingly.
In this new version we inserted 2 new Tables to better define advantages of new 3D tools. However since this Review refers to 3DTTE we underlined in Table 2 that while 3DTEE has a very important role in the interventional field (and it may be mandatory in several procedures) 3DTTE has not nowadays clear indications in this field.

Round 2
Reviewer 1 Report
Thank you. All my comments have been appropriately addressed. I have no further comments.
Reviewer 2 Report
The authors have addressed my comments. I am inclined to recommend acceptance of the revised manuscript.